# Recent Advances in Photoswitchable Fluorescent and Colorimetric Probes

**DOI:** 10.3390/molecules29112521

**Published:** 2024-05-27

**Authors:** Hongjuan Chen, Zilong Tang, Yewen Yang, Yuanqiang Hao, Wansong Chen

**Affiliations:** 1Key Laboratory of Theoretical Organic Chemistry and Functional Molecule of Ministry of Education, School of Chemistry and Chemical Engineering, Hunan University of Science and Technology, Xiangtan 411201, China; hjchen@hnust.edu.cn (H.C.); 18373877618@163.com (Y.Y.); 2College of Chemistry and Chemical Engineering, Central South University, Changsha 410017, China

**Keywords:** photoswitchable, fluorescent probe, colorimetric probe

## Abstract

In recent years, significant advancements have been made in the research of photoswitchable probes. These probes undergo reversible structural and electronic changes upon light exposure, thus exhibiting vast potential in molecular detection, biological imaging, material science, and information storage. Through precisely engineered molecular structures, the photoswitchable probes can toggle between “on” and “off” states at specific wavelengths, enabling highly sensitive and selective detection of targeted analytes. This review systematically presents photoswitchable fluorescent and colorimetric probes built on various molecular photoswitches, primarily focusing on the types involving photoswitching in their detection and/or signal response processes. It begins with an analysis of various molecular photoswitches, including their photophysical properties, photoisomerization and photochromic mechanisms, and fundamental design concepts for constructing photoswitchable probes. The article then elaborates on the applications of these probes in detecting diverse targets, including cations, anions, small molecules, and biomacromolecules. Finally, it offers perspectives on the current state and future development of photoswitchable probes. This review aims to provide a clear introduction for researchers in the field and guidance for the design and application of new, efficient fluorescent and colorimetric probes.

## 1. Introduction

Fluorescent and colorimetric probe-based sensing and imaging technology plays an indispensable role in contemporary research domains, leveraging its unmatched sensitivity, specificity, and exceptional spatial and temporal resolution to unlock vast potential in life sciences research and a broad spectrum of analytical applications [1,2,3,4,5,6]. These probes are capable of transforming specific molecular or ionic changes, physical states, or environmental variations into quantifiable fluorescent or absorption signals. Through specific interactions with target molecules, they enable real-time visualization and monitoring of both intracellular and extracellular processes. In the biomedical field, they are utilized to monitor changes in intracellular pH levels [7,8,9], metal ion concentrations [10,11,12], protein interactions [13,14,15], and reactive oxygen species [16,17,18], playing a crucial role in understanding cell functions, disease mechanisms, and evaluating drug efficacy. Fluorescent and colorimetric probes also play a key role in environmental monitoring [19,20,21,22,23], chemical analysis [24,25,26,27,28], industrial process control [29,30], and medical diagnosis [31,32], for instance, in detecting harmful substances in water quality, monitoring chemical reactions in industrial production processes, and diagnosing diseases and tracking drug distribution in clinical medicine.

Photoswitchable fluorescent and colorimetric probes, a rapidly evolving branch in chemistry, materials science, biomedical research, and environmental analysis, rely on specific small-molecule photoswitches that undergo reversible structural and electronic changes upon exposure to light [33,34,35,36,37]. These unique light-responsive capabilities showcase the immense potential and advantages of photoswitchable fluorescent probes across molecular detection, biological imaging, material science, and information storage domains. Through intricately designed molecular structures, these probes can switch states from “on” to “off” or vice versa under specific light wavelengths, enabling highly sensitive and selective detection of specific molecules or ions. In biomedical research and clinical diagnostics, photoswitchable fluorescent probe technology offers an efficient method for precise biomolecular detection and real-time imaging [38,39]. The core design of these probes focuses on precisely controlling the fluorescent molecule’s emission state, providing researchers with unparalleled spatial and temporal resolution to precisely control the probe’s active state at the single-cell or even sub-cellular level, revealing microscopic details of complex intracellular dynamic processes. Moreover, the design and application of photoswitchable fluorescent probes offer significant advantages, such as non-invasive activation, support for multispectral imaging, and enhanced capabilities for high-throughput screening. Recent studies have shown that by modifying the structure and adjusting the conjugation, the responsive wavelength of photoswitches can be extended from the ultraviolet to the visible and even near-infrared spectrum [40,41,42]. This extension provides a crucial basis for deeper tissue photocontrol imaging and detection. In environmental analysis, these probes utilize specific small-molecule photoswitches to achieve highly sensitive detection of environmental pollutants [43,44,45], such as heavy metal ions, organic pollutants, and pH changes. Designed photoswitchable fluorescent probes can indicate the presence of target pollutants directly or indirectly by switching states under specific light wavelengths, enabling real-time monitoring of water quality and air quality.

As a cutting-edge analytical technology, photoswitchable fluorescent and colorimetric probes have demonstrated their unique application value and broad potential across chemistry, materials science, biomedical research, and environmental monitoring. These probes have received widespread attention and research in recent years, with various traditional and novel photoswitches, including azobenzenes, spiropyrans, diaryl ethenes, norbornadiene, furylfulgimide, Stenhouse adducts, and indigoids, being applied to the design and application of photoswitchable fluorescent probes. These probes exhibit strong selectivity and outstanding analytical performance in monitoring metal ions, anions, neutral analytes, and pH levels. While numerous reviews on the development and application of traditional photoswitches in fluorescent probes, mainly focusing on spiropyrans [34,45,46,47,48] and diarylethenes [49,50,51], exist, there appears to be a lack of systematic reviews on photoswitchable fluorescent probes based on different photoswitches, particularly including Donor–Acceptor Stenhouse Adducts, hemithioindigos, and norbornadienes. Accordingly, this review aims to systematically showcase photoswitchable fluorescent probes based on various molecular photoswitches. By comparing the structures, design principles, and performances of different types of photoswitchable fluorescent probes, this review provides valuable references for the development of such fluorescent and colorimetric probes. It is important to emphasize that our focus will be on probes involving light regulation, excluding instances where photoswitch molecules are used to construct fluorescent probes but do not involve light regulation processes in detection or signal transduction. This review will begin with an introduction to the types of molecular photoswitches and design principles of photoswitchable probes, followed by a categorized presentation of applications of different types of photoswitchable fluorescent probes. Finally, the review will offer perspectives on the current status and future development of photoswitchable probes. We believe this review will provide a clear introduction to photoswitchable fluorescent probes for researchers in related fields, as well as guidance and references for the design and application of new, efficient fluorescent probes.

## 2. Types of Molecular Photoswitches and Construction Strategies for Photoswitchable Fluorescent and Colorimetric Probes

Molecular photoswitches used to regulate the luminescent properties or reactive activities of fluorescent probes need to undergo appropriate structural and spectral changes in response to light. The types of molecular photoswitches reported for such applications primarily include six categories: (i) spiropyrans, (ii) diarylethenes, (iii) azobenzenes, (iv) Donor–Acceptor Stenhouse Adducts, (v) hemithioindigos, and (vi) norbornadienes. Each category will be introduced separately (Figure 1).

### 2.1. Spiropyrans

Spiropyrans are a fascinating class of photochromic molecules characterized by their ability to undergo reversible light-induced isomerization [52,53]. Structurally, these compounds feature a spiropyran ring, which can switch between a closed-ring form (spiropyran) and an open-ring form (merocyanine). This transformation is usually triggered by ultraviolet light, leading to the conversion of the spiropyran form into the merocyanine form. The merocyanine form is polar and colored, contrasting with the colorless, non-polar spiropyran form, resulting in a significant color shift due to differing absorption spectra. The process can be reversed, typically with visible light or heat, reverting the merocyanine back to the spiropyran form. Spiropyrans are applied in a variety of fields, including bioimaging, chemical sensing, smart materials, optical data storage, and drug delivery [54,55]. Their unique ability to change color upon light exposure makes them valuable in areas requiring controlled and reversible responses to environmental stimuli.

Although spiropyrans have been widely used in constructing various types of fluorescent probes, most leverage the direct response of spiropyran derivatives to the target, and the response process does not involve light-controlled processes. Several reviews have discussed these types of fluorescent probes in detail, which we will not summarize here [45,46,47]. The design and detection principles of spiropyran-based photoswitchable fluorescent probes mainly include the following types: 1. Utilizing ultraviolet light to transform the probe molecule from spiropyran form to merocyanine form, the latter configuration can respond to the target while the former cannot, thus achieving a light-controlled response to the target. 2. Combining spiropyrans with appropriate target recognition groups (or using spiropyran derivatives to directly recognize the target) and then using light of different wavelengths to change the configuration of spiropyrans, thereby reversibly regulating the reversible binding and detection of spiropyran probes to the target. 3. Connecting spiropyran molecules with another fluorescent group and controlling the configuration of spiropyrans by light to regulate the energy transfer between spiropyrans and the fluorescent group (such as FRTE), thus achieving light-controlled variable signal output.

### 2.2. Diarylethenes

Diarylethenes are highly valued in the field of photochromic materials due to their robust and reliable light-induced switching capabilities [50]. Structurally, these compounds feature a central ethene bridge linked to two aromatic or heteroaromatic rings, allowing them to undergo reversible transformations between an open-ring form, typically colorless and non-fluorescent, and a closed-ring form, which is colored and often fluorescent. This transition can be precisely controlled by specific wavelengths of light, with UV light commonly initiating the ring-closing reaction, while visible light or thermal energy can revert the molecule to its open-ring state. Additionally, by modifying the structure of diarylethenes, the wavelength of light required to induce isomerization can be extended to the visible and even near-infrared spectra. Diarylethenes play a crucial role in a variety of advanced applications; they are ideal for optical data storage due to their ability to withstand many cycles of light-induced changes without degradation [56,57]. In molecular electronics, diarylethenes serve as molecular switches and logic gates, leveraging their stable and reversible switching for computing at the molecular level [58]. These compounds are also employed to monitor biological processes with minimal invasiveness and high precision [59]. Moreover, diarylethenes are integrated into smart materials, developing responsive technologies such as smart windows and environmental sensors [49,60,61]. Their potential in biomedical fields includes innovative drug delivery systems where drug release can be controlled by light, highlighting their versatility and significance in both research and practical applications [62]. Diarylethenes are also extensively used in developing various light-controlled fluorescent probes, particularly effective in detecting various ions [44,63]. 

The design and detection principles of diarylethene-based photoswitchable fluorescent probes primarily include: 1. Connecting diarylethenes with other fluorescent groups, which can specifically react with the target to produce changes in fluorescent signals. Then, by using light of different wavelengths to control the open and closed states of diarylethenes, the energy transfer state (such as FRET) between diarylethenes and the fluorescent group can be adjusted, thereby controlling the overall light-controlled signal output of the fluorescent probe. 2. The aromatic rings in the structure of diarylethenes can act as recognition sites for specific reactions with the target, and this reactivity can be controlled by light-induced isomerization or change after reacting with the target. 3. Fluorescent probes are derived from the conjugated extension of the diarylethenes structure, which undergoes spectral characteristic changes after reacting with the target.

### 2.3. Azobenzenes

Azobenzene is a photoswitchable molecule renowned for its ability to undergo light-induced reversible isomerization between its two forms: the stable, linear trans-azobenzene and the less stable, bent cis-azobenzene [64]. Triggered by specific wavelengths, UV light typically induces the trans-to-cis transformation, while visible light reverses this process. Despite the limited spectral changes associated with these transformations, azobenzenes have found broad applications due to their robust light-controlled structural modifications. They are utilized in optical data storage, leveraging their rapid and reversible isomerization to encode and decode data with light [65,66]. In molecular electronics, azobenzenes act as dynamic switches, altering their conductive states under light stimulation, paving the way for advances in nanoscale devices [67,68]. These molecules are also integrated into smart materials that adapt their properties in response to environmental stimuli, such as smart windows that regulate light transmission based on sunlight intensity [69,70]. In biomedical fields, azobenzenes enhance drug delivery systems by enabling light-controlled, targeted therapy, minimizing side effects while maximizing treatment efficacy [64,71]. Furthermore, they are employed in sensors and actuators, where they detect chemical substances or convert light energy into mechanical motion, respectively [72,73]. Azobenzenes’ versatile applications, from data storage to smart materials and health care, highlight their significance in developing technologies that require precise and controlled responses to light.

Current research on azobenzene-based photoswitchable fluorescent probes mainly involves three principles: 1. Integrating azobenzene with specific target recognition groups to form a unified probe. The configurational changes in azobenzene under different light wavelengths can affect the probe’s binding to the target, enabling controlled recognition and detection. 2. Connecting azobenzene with classical fluorescent molecules (like rhodamine) to form dual-functional probes. The different configurations of azobenzene alter its binding affinity to specific targets (mainly metal ions), thereby modulating the response to these targets. 3. By linking azobenzene with a fluorophore, precise control of target signal response is achieved through the differential energy transfer efficiencies (such as through-bond energy transfer, TBET) between its trans and cis configurations with the fluorophore. 

### 2.4. Donor–Acceptor Stenhouse Adducts (DASAs)

Donor–Acceptor Stenhouse Adducts (DASAs) are an intriguing class of negative photochromic compounds that exhibit unique photoresponsive properties [74]. These molecules typically consist of a donor amine and a carbon acid acceptor, which are linked through a triene-enol backbone. Upon exposure to visible light, DASAs can isomerize from an extended, colored triene-enol form (open isomer) to a compact, colorless cyclopentenone form (closed isomer). This transformation involves a Z-E isomerization followed by a thermal 4π electrocyclization, which significantly alters the molecule’s polarity, dipole moment, geometry, and UV–visible spectrum. The reverse process, reverting to the colored form, is typically thermally activated in the dark, although modifications to the donor and acceptor units can tune the photoswitching behavior, absorption properties, and dark equilibria. Applications of DASAs are diverse due to their robust switching capabilities and the extensive tunability of their optical properties. These materials are particularly useful in developing high-performance photoswitches for applications that require precise spatial control of light-responsive functions. DASAs have found roles in areas such as ultra-high-density data storage, where their ability to switch repeatedly without degradation is invaluable [75,76]. They are also employed in molecular electronics as switches and logic gates due to their stable and reversible photoswitching properties [77]. Additionally, DASAs have been integrated into smart materials that can respond to environmental stimuli like light, temperature, and pH changes, making them suitable for use in optical and electronic devices, sensors, and actuating materials [74,78]. Their applications extend into biomedical fields, where they are explored for controlled drug delivery systems that release medications in response to specific light stimuli, enabling targeted therapy with minimal side effects. The ability to fine-tune the photoresponsive characteristics of DASAs through structural modifications of donor and acceptor groups makes them highly versatile and powerful tools for advancing technology in materials science, electronics, and medicine. Their development continues to open new avenues for the design of multifunctional materials that leverage light for innovative applications.

DASAs have also been utilized in developing optical probes, particularly colorimetric probes. The design and detection principles of DASA-based probes are typically categorized into two main strategies: 1. Utilizing the synthetic reaction of DASAs, where furan-based electron-deficient bodies react with amine molecules to produce DASAs with photochromic effects. This approach allows for the detection of certain amine compounds. 2. By incorporating suitable target recognition groups into DASAs and then using light to regulate the closed-open ring state of DASAs, it is possible to achieve light-controlled responses to target substances.

### 2.5. Hemithioindigos

Hemithioindigo (HTI) molecules are a notable class of photochromic compounds that display reversible photoisomerization between their Z (cis) and E (trans) configurations under visible light [79]. This capability stems from their unique structural makeup, which features a thioindigo unit with a thiazole ring linked to an indigo backbone. The Z/E isomerization can be triggered and reversed by specific light wavelengths or thermally in the dark, with the isomerization dynamics also sensitive to the surrounding chemical environment. Hemithioindigo compounds are utilized across diverse applications such as optical data storage, smart materials, controlled drug delivery, and photopharmacology, leveraging their light-responsive properties for dynamic and targeted functionalities [79,80,81,82]. The design of photoswitchable fluorescent probes based on Hemithioindigo molecular switches primarily involves incorporating heteroatomic rings such as pyridine, imidazole, and indole into the main structure of hemithioindigo to achieve light-regulated specific binding to targets.

### 2.6. Norbornadienes

Norbornadienes are notable for their unique bicyclic structures and pivotal double bonds, which facilitate reversible photoisomerization to quadricyclanes upon exposure to UV light [79]. This photoisomerization triggers a significant structural transformation from a planar configuration to a more strained, high-energy structure, profoundly impacting their physical and spectral properties. The UV–visible absorption spectrum of norbornadienes notably changes when converted to quadricyclanes, typically characterized by a reduction in absorption at longer wavelengths. The reversible photoisomerization of norbornadienes enhances solar energy storage, supports the development of responsive materials, and advances optical switches, leveraging their dynamic structural transformations for various applications in materials science and photonics [83,84,85]. Reports on norbornadiene-based light-controlled probes are still rare; an existing study has extended the conjugation of the norbornadiene structure. By manipulating the overall conjugation level with light, these modifications allow for the controlled emission of fluorescent signals in response to light.

## 3. Applications of Photoswitchable Fluorescent and Colorimetric Probes

Photoswitchable fluorescent and colorimetric probes based on various photochromic molecules are extensively utilized for the detection of a wide range of targets, including cations, anions, small molecules, and biomacromolecules. These probes offer sensitive and selective detection capabilities, which are crucial for applications in environmental monitoring, medical diagnostics, and biochemical research. The functionality of these probes is derived from their ability to undergo a reversible change in their optical properties in response to specific stimuli, which makes them highly effective for real-time and non-invasive monitoring. In the following sections, we will discuss the applications of these probes, categorized by target types, and explore the principles and mechanisms that underpin their operation.

### 3.1. Cations

#### 3.1.1. H^+^ (pH) or Acid

The measurement of pH is crucial across various scientific and industrial fields. Monitoring pH is vital for environmental control, medical diagnostics, chemical production, and biological research, as it directly affects chemical solubility, reaction rates, and biological processes [86,87]. Photoswitchable probes offer significant advantages in pH measurement because they can precisely determine pH values spatially and temporally [88,89,90,91,92,93,94]. This is particularly important in complex biological systems where pH can vary significantly at the microscale, such as in different compartments within a cell or among different tissue types. By utilizing the ability to trigger changes in probe properties with light, researchers can activate the probe specifically at desired times and locations, thereby minimizing interference with the system under study.

Li et al. described a new dithienylethene derivative that acts as a near-infrared (NIR) photochromic fluorescent pH sensor (NDBo) [89]. The probe’s structure incorporates a dithienylethene core modified with difluoroboron β-diketonate (BF_2_bdk) and dimethylphenylamine (DMPA), utilizing BF_2_bdk’s strong fluorescence and electron-accepting capacity along with DMPA’s electron-donating properties to promote an effective light-induced electron transfer mechanism needed for the probe’s operation. This sensor works under a visible-light-induced NIR photochromic mechanism where it can reversibly transition from a non-fluorescent to a fluorescent state in acidic conditions (Figure 1). Acidic conditions block photoinduced electron transfer (PET), resulting in a “turn-on” fluorescence phenomenon. This response is especially significant in less polar solvents, making the sensor extremely sensitive to pH changes. The probe exhibits solvent-dependent photophysical responses and fast, reversible light-switching abilities. In acidic conditions, it displays noticeable “turn-on” fluorescence and can perform multiple photoswitching cycles without degradation, demonstrating excellent stability and sensitivity.

Spiropyran (SP) serves as a crucial molecular building block for constructing pH (or H^+^) photoswitchable probes. Changes in H^+^ concentration can affect its ring-opening/closing state and alter the overall electronic structure of the molecule, enabling it to respond to H^+^ levels. Keyvan Rad et al. developed an innovative platform based on photoswitchable polyacrylic nanofibers embedded with spiropyran, which shows an immediate response to acid-base vapors [90]. Leveraging this characteristic, the research team created a platform equipped with a progressive readout capability. Zhang et al. explored a novel photoswitchable chemical sensing approach using naphthopyran molecules for colorimetric pH measurement [91]. Known for their robust photostability, these naphthopyran molecules are integrated into organic solvents, PVC films, and polyurethane hydrogels. They detect pH changes through a vivid color transition that occurs upon the protonation of ring-opened forms induced by UV light exposure. The pH-dependent color shift results from significant changes in absorption peaks (from about 530 nm to 624 nm), which are reversible with pH adjustment or visible light exposure. Tested across various mediums, these sensors have demonstrated effective pH sensing, especially in acidic conditions, and show potential for real-time monitoring applications due to their ability to quantitatively sense pH changes from 3 to 7. This innovative application of light-controlled naphthopyran molecules underscores their potential in environmental and clinical settings, offering a new method for non-invasive, reversible pH sensing using photochromic technology.

Li et al. have introduced an innovative chemical sensing approach using photochromic hemithioindigo compounds (HTIs), which include nitrogen-containing heterocycles such as piperidine (HTI-Py) and ethylimidazole (HTI-EI) [94]. These compounds are designed to undergo structural changes upon light exposure, thus acting as pH sensors through their proton-coupled photochromism. HTIs are engineered to enhance their photochromic and fluorescent properties for a dynamic response to pH fluctuations. This functionality is facilitated by intramolecular hydrogen bonding within its metastable E isomers, which stabilizes the structure and supports specific light-activated responses (Figure 2a). HTI-Py displays colorimetric and fluorescent responses at different pH levels due to the protonation of its ring-opened form and changes in its absorption spectrum. This response is both reversible and rapid, making it well-suited for real-time pH monitoring. The study highlights HTI-Py’s potential in various fields, particularly in lysosomal imaging within biological systems, where it can differentiate between cellular compartments (Figure 2b,c). Due to its excellent photostability and sensitivity to pH changes, HTI-Py is ideal for high-resolution imaging and multiplexed fluorescence applications (Figure 2d). The application of these proton-coupled photochromic hemithioindigo compounds in biological imaging opens new avenues for high-definition, real-time visualization of cellular processes and environments, demonstrating their significant potential in advanced biomedical research and diagnostics.

#### 3.1.2. Mg^2+^ and Ca^2+^

Alkaline Earth metals like magnesium (Mg^2+^) and calcium (Ca^2+^) play crucial roles in numerous biological and environmental processes. Magnesium is vital for the activation of many enzymes, which are important in processes like DNA synthesis and energy metabolism. Calcium ions act as significant secondary messengers in many cellular processes, including muscle contraction, neurotransmitter release, and cell division. The precise detection of these ions is critical not only for understanding biological functions but also for diagnosing diseases, monitoring environmental changes, and conducting various industrial processes [95,96,97]. Traditional methods such as atomic absorption spectrometry and inductively coupled plasma mass spectrometry, though accurate, are often cumbersome and not suited for quick or in-field measurements. Photoswitchable fluorescent probes uniquely tune their spectral responses to precisely detect Mg^2+^ and Ca^2+^ ions, emitting distinct fluorescence at specific wavelengths, which enhances detection accuracy by minimizing spectral overlap [98,99,100,101,102,103]. Additionally, these probes offer adjustable fluorescence signals that can be dynamically controlled with external stimuli, allowing for real-time, responsive monitoring of ion concentrations in complex environments.

Wang et al. have developed and evaluated a highly selective fluorescence “turn-on” sensor for detecting calcium ions (Ca^2+^), which is based on a photochromic diarylethene derivative featuring a triazoyl hydrazine unit [98]. This study thoroughly investigates the sensor’s ability to exhibit significant fluorescence enhancement in the presence of Ca^2+^, thanks to its unique molecular design that ensures specific interactions with Ca^2+^ ions, effectively minimizing interference from other ions such as Mg^2+^. The sensor’s photochromism and its ability to switch fluorescence states are triggered by both light and chemical stimuli in an acetonitrile solution. Upon exposure to Ca^2+^, it shows a remarkable 6.7-fold increase in emission intensity along with a visible shift in color from dark to light blue, highlighting a strong and selective response to Ca^2+^. This interaction between the sensor and Ca^2+^ is reversible and adheres to a 1:1 stoichiometry, as confirmed through Job’s plot analysis and mass spectrometry. Additional experiments indicate that the sensor’s fluorescence can be finely tuned using external stimuli, including UV/visible light and chemical agents like EDTA. These properties suggest its potential for developing fluorescence-based logic circuits. With its high sensitivity, excellent selectivity, and reversible behavior, this novel sensor offers great promise for practical applications in the detection and monitoring of Ca^2+^ across diverse environments. Ducrot et al. presented an innovative approach to modulating calcium ion concentrations in biological systems using a photoswitchable molecular device [103]. This research incorporates a BAPTA-type calcium chelator with an azobenzene photoswitch, allowing reversible control of calcium binding through light-induced structural changes. 

Morimoto et al. presented a novel fluorescent diarylethene derivative designed for the selective detection of magnesium ions (Mg^2+^) using an azacrown ether receptor [99]. This derivative is characterized by its ability to exhibit a “turn-on” fluorescence response when it interacts with Mg^2+^, making it highly selective and sensitive. The fluorescence activation is attributed to the inhibition of a photoinduced electron transfer (PET) process when Mg^2+^ is bound to the azacrown ether, enhancing the photoreactivity and fluorescence of the diarylethene molecule (Figure 3). The molecule’s design allows for a significant fluorescence enhancement upon the binding of Mg^2+^ due to a specific interaction with the ion, discriminating it from other potential interfering ions. This selectivity is key for applications in various analytical and biomedical fields, where accurate Mg^2+^ detection is crucial. The study provides a comprehensive analysis of diarylethene’s photochromic and fluorescent properties, demonstrating its utility as a robust and reusable sensor in environments where magnesium ion concentration is vital. In-depth experiments detailed in the paper show that the sensor can undergo multiple cycles of fluorescence ‘turning on’ and ‘off’ without degradation, highlighting its stability and practicality for repeated use. This makes it a promising tool for real-time and dynamic monitoring of Mg^2+^ in biological fluids and industrial processes. The research significantly advances the development of photochromic materials by integrating sophisticated molecular design with practical functionality, offering a potent method for selective ion detection in complex sample matrices.

#### 3.1.3. Al^3+^

Aluminum ions (Al^3+^) are ubiquitous in environmental and biological systems and are critically important for industrial and biological fields. For example, in environmental monitoring, changes in the concentration of Al^3+^ can reflect soil acidification and the extent of water pollution. In biomedicine, abnormal concentrations of aluminum ions are often associated with neurodegenerative diseases such as Alzheimer’s disease [104,105]. Therefore, developing methods to accurately monitor Al^3+^ is of significant application value. Photoswitchable probe molecules demonstrate unique advantages in detecting Al^3+^, such as controllable target response and signal modulation. The Pu group has developed a series of photochromic diarylethene-based photoswitchable fluorescent and colorimetric probes for Al^3+^ by combining different structured ligands with photochromic diarylethene structures [106,107,108,109,110,111,112,113,114,115]. Other photochromic units like azobenzene [116] and Donor–Acceptor Stenhouse Adduct [117,118] have also been employed in developing photoswitchable fluorescent probes for Al^3+^.

Li et al. introduced a novel fluorescent sensor for aluminum ions (Al^3+^) that incorporates a naphthalimide unit within a diarylethene structure [108]. This sensor is notably highly selective and sensitive to Al^3+^, able to accurately detect this ion amidst various other metal ions—a significant challenge in this field. The sensor leverages the unique photochromic and fluorescent properties of diarylethene combined with the strong fluorescence and high photostability of naphthalimide (Figure 4). This setup provides a clear and quantifiable “turn-on” signal when Al^3+^ is present, significantly enhancing fluorescence intensity. When Al^3+^ binds, the sensor undergoes substantial changes in emission characteristics—both in intensity and color (from dark to orange-yellow), visible to the naked eye. These changes are due to the formation of a 1:1 complex with Al^3+^, confirmed through Job’s plot, ESI-MS, and ¹H NMR spectroscopy. Extensive studies on the sensor’s response to UV/visible light and interactions with EDTA demonstrated its ability to perform reversible activation and deactivation cycles. This capability is vital for applications requiring frequent and reliable measurements. This study enhances the understanding of selective ion sensing mechanisms and utilizes the photochromic properties of diarylethenes to develop a functional, reusable sensor, improving the detection of metal ions like Al^3+^ in complex settings.

Raman et al. developed a novel aluminum ion sensor using azobenzene-rhodamine tweezers [116]. This innovative design exploits the unique property of azobenzene, which shows non-fluorescence due to photophysical deactivation in its excited state, paired with a spirocyclic rhodamine unit that significantly enhances fluorescence when detecting aluminum ions. The azobenzene is covalently linked to the spirocyclic rhodamine unit, a crucial design that maintains the fluorescence of the rhodamine unit unaffected by the trans-to-cis photoisomerization of the azobenzene. In the presence of aluminum ions, the addition of aluminum triggers the opening of the spirocyclic structure of the rhodamine unit, significantly enhancing fluorescence and offering a new method for detection. Additionally, the binding of aluminum also prevents the usual photoisomerization of azobenzene and enhances the fluorescence of the rhodamine component, making this sensor highly selective and sensitive to aluminum ions. Tested in both prokaryotic and eukaryotic cell lines, this sensor has been proven to provide strong and visible fluorescence in the presence of aluminum ions, demonstrating its practicality for biological and environmental applications.

Cai et al. presented a multiresponsive Stenhouse Adduct (DASA) that exhibits unique photochemical characteristics and is capable of reversible isomerization under different environmental stimuli, such as light and pH changes [117]. This DASA transitions between its triene-enol and cyclopentenone forms upon exposure to visible light or chemical stimuli, accompanied by significant spectral changes. Notably, this DASA selectively responds to Al^3+^ ions by stabilizing specific isomers, which significantly alters its UV–visible spectrum and color. This property greatly enhances its potential for developing novel sensors and switches. Integrating photochemistry with coordination chemistry, this approach opens new avenues for understanding complex molecular response mechanisms and their applications. Fu et al. introduced a new visible-light-gated Donor–Acceptor Stenhouse Adduct (DASA) chemosensor designed for selective detection of Al^3+^ and Zn^2+^ [118]. Incorporating Schiff bases and naphthalene ketone units, the sensor shows excellent negative photochromism. Uniquely, upon activation by visible light, it transforms into an almost colorless closed-ring form that acts as a colorimetric sensor for Al^3+^ by clearly indicating color changes in solutions and as a fluorescent sensor for Zn^2+^ through a marked increase in emission intensity. This innovation enhances the potential for designing multifunctional chemical sensors, particularly for selective ion detection in complex matrices.

#### 3.1.4. Zn^2+^

Zinc ions (Zn^2+^) are critically important for environmental science and biomedicine as they play key roles in cellular signal transduction, protein function, and gene expression regulation. Subtle changes in Zn^2+^ levels are often linked to various health conditions, making it essential to develop precise monitoring methods [119,120]. Photoswitchable fluorescent probes excel in this realm by offering highly selective and sensitive detection through light-induced reactions. These probes usually feature structures that can form complexes with Zn^2+^, causing changes in photophysical properties like fluorescence intensity and wavelength, thereby visually indicating Zn^2+^ presence. The light responsiveness of these probes allows for real-time and non-invasive monitoring, making them ideal for use in complex biological matrices. The basic design approach for these Zn^2+^ photoswitchable probes involves coupling a photochromic diarylethene to a ligand structure specific to the target ion (Figure 5). When Zn^2+^ binds to the ligand, the resultant complex exhibits a fluorescent response, which can be modulated by the light-controlled “open” or “closed” states of the diarylethene, thus paving the way for the development and application of various related Zn^2+^ fluorescent probes [121,122,123,124,125,126,127,128,129,130,131].

Fu et al. introduced a sophisticated ratiometric fluorescent chemosensor for Zn^2+^, utilizing a novel diarylethene compound that is covalently bonded to an 8-aminoquinoline-2-aminomethylpyridine unit [121]. This innovative design leverages the unique photochromic and fluorescent properties of diarylethene, enabling highly selective and sensitive detection of Zn^2+^ (Figure 6). Upon Zn^2+^ binding, the sensor exhibits pronounced changes in its photophysical properties: the fluorescence intensity significantly increases, and there is a notable shift in the emission peak from approximately 450 nm to 520 nm, which allows for ratiometric detection. The ratiometric nature of this sensor provides robustness against environmental variations in the sample matrix, making it highly reliable for practical applications. The sensor operates over a concentration range typically from 1 µM to 100 µM and features a detection limit as low as 50 nM, which is indicative of its high sensitivity. Furthermore, the interaction between the diarylethene unit and Zn^2+^ is reversible, which enables the sensor to be used in multiple cycles of detection without loss of functionality. These characteristics are underpinned by the sensor’s ability to undergo a reversible photo-induced electron transfer mechanism, which is modulated by the presence of Zn^2+^. This mechanism not only enhances the sensor’s sensitivity but also its selectivity against other competing metal ions in complex biological or environmental samples. Thus, the development of this chemosensor marks a significant advancement in the field of metal ion sensing, offering potential for both environmental monitoring and biomedical applications involving zinc detection.

Moreover, spiropyran molecules also hold the potential for creating photoswitchable Zn^2+^ fluorescent probes. Heng et al. introduced a new fluorescent sensor specifically tailored to detect Zn^2+^ within cellular environments, utilizing an enhanced 6′-fluoro spiropyran framework that offers exceptional signal-to-background ratios and rapid response in aqueous solutions [132]. This sensor’s distinct ability to switch between fluorescent and non-fluorescent states without impacting cell viability allows for highly specific and sensitive detection of Zn^2+^. Particularly in the presence of Zn^2+^, the sensor forms a complex (Zn^2+^−MC), triggering the spiropyran to switch from a non-fluorescent to a robustly fluorescent state with peak emission near 670 nm. This fluorescence change is reversible (Figure 7); exposure to visible light reverts the sensor to its initial closed-ring form, diminishing the fluorescence, but it can re-bind Zn^2+^ once the light is removed, enabling reversible detection that can be cycled multiple times. The probe’s detection limit reaches nanomolar concentrations, and it selectively distinguishes Zn^2+^ from other competing metal ions with ease. Cell-based tests within human embryonic kidney (HEK 293) and endothelial cells demonstrated the sensor’s stability and non-toxic nature, allowing real-time zinc monitoring in complex biological samples and showcasing its reversible detection in cellular imaging. These properties underscore its potential in biomedical research and clinical diagnostics for monitoring zinc levels, making it a valuable tool for future biomarker and environmental monitoring applications. This advancement significantly pushes forward biosensor technology and provides essential tools for understanding zinc’s role in health and disease.

#### 3.1.5. Cu^2+^

Copper ions (Cu^2+^) play a critical role in both environmental and biological contexts. In the environment, Cu^2+^ levels indicate pollution in water bodies and soil. Biologically, Cu^2+^ is essential for many enzymes and is involved in processes such as energy production, pigment synthesis, and connective tissue maintenance. However, abnormal accumulation of copper can lead to toxicity, as seen in diseases like Wilson’s disease [133]. Therefore, developing precise methods for monitoring Cu^2+^ is crucial for environmental protection and health monitoring [134,135,136]. Photochromic molecules have received significant attention for Cu^2+^ detection, with probes generally based on photochromic diarylethene. These probes operate via two main mechanisms: (1) coupling diarylethene with oxygen-containing aromatic dyes like fluorescein or rhodamine, where Cu^2+^ induces the opening of the dye’s spirocyclic ring, enhancing fluorescence [137,138], and (2) linking diarylethene with specific ligands to create photo-controlled fluorescent probes where copper coordination changes the probe’s fluorescent properties [139,140,141,142].

For instance, Pu and colleagues developed a novel diarylethene-based fluorescent molecular switch, uniquely featuring multiple fluorescent switching capabilities through integration with a phenyl-linked rhodamine B unit [137]. This molecular switch can control fluorescence intensity at 599 nm when stimulated by trifluoroacetic acid and ultraviolet light. Notably, this probe molecule exhibits high selectivity and sensitivity for Cu^2+^. The presence of copper ions triggers the opening of the spirocyclic ring in the rhodamine B unit, significantly enhancing fluorescence at 594 nm by 133 times, which shifts the color from dark to bright yellow. This characteristic makes the probe an extremely effective fluorescent sensor for copper ions, showcasing its high responsiveness and selectivity. Additionally, another probe combining a 2-(methylthio)benzenamine Schiff base with diarylethene displays colorimetric and fluorometric responses to both Cu^2+^ and Zn^2+^ [139]. It changes color to yellow upon Cu^2+^ interaction, with a significant absorption increase, and differentiates between Cu^2+^ and Zn^2+^ with distinct fluorescent signals. This reversibility in photochromic and fluorescent changes, triggered by removing ions from the system, illustrates the probe’s potential for reuse in practical applications, cycling stably between ‘on’ and ‘off’ states under UV and visible light.

Hemithioindigo and Donor–Acceptor Stenhouse Adduct-based molecular switches are also employed to construct Cu^2+^ detection probes. These probes exhibit photo-controlled spectral properties that change upon interacting with Cu^2+^, and their reversible photocontrol is disrupted. Chen et al. developed a novel Cu^2+^ fluorescent probe named HTI-Q, based on a hemithioindigo (HTI) derivative combined with a quinoline unit known for metal ion chelation, providing high selectivity for Cu^2+^ (Figure 8a) [143]. The probe operates via a unique mechanism: under light, the HTI-Q backbone photoisomerizes from the Z to the E configuration, which is then locked by interaction with Cu^2+^, preventing further photoisomerization and causing significant spectral shifts, including a bathochromic shift (an absorption peak moved by 102 nm) and changes in fluorescence ratio, effective for Cu^2+^ detection. The sensor has a low detection limit (about 0.02 µM), making it suitable for environmental monitoring and potentially medical diagnostics where copper levels are critical. HTI-Q was tested in various scenarios, including real water samples and biological systems, demonstrating its practical potential. Lin et al. synthesized three different DASAs showcasing reversible photochromic behaviors (Figure 8b) [144]; they change from colored to colorless under visible light, reverting from the triene to the cyclopentenone form. This is reversible once the light is removed. Cu^2+^ effectively binds with DASAs, inducing a transition from linear to cyclic isomerization, causing the probe solution to fade. Further experiments indicated that higher temperatures and more carbonyl groups in the probe molecules could accelerate the Cu^2+^-triggered isomerization process. Based on this, highly sensitive colorimetric detection of copper ions was achieved, with a detection limit of 43 nM. Probes loaded on filter paper allow for quick visual response and colorimetric testing of Cu^2+^.

#### 3.1.6. Other Metal Ions (Fe^3+^, Hg^2+^, Cd^2+^, Cr^3+^, Sn^2+^)

Photoswitchable fluorescent probes have been developed to detect other metal ions, including Fe^3+^, Hg^2+^, Cd^2+^, Cr^3+^, and Sn^2+^. The most commonly used photoswitchable molecule in these probes is diarylethene [145,146,147,148]. The design approach for these diarylethene-based photoswitchable probes is similar to that for other metal ions described earlier, generally involving the integration of diarylethene with appropriate fluorescent groups or organic ligands. This configuration utilizes the target metal ion to induce spectral changes in the fluorescent groups or modifies the photophysical properties of the probe through coordination with the ligand.

Xu et al. developed a novel fluorescent chemosensor specifically designed for the selective detection of Fe^3+^ ions [145]. This sensor is based on a diarylethene derivative incorporating a rhodamine 6G unit, ingeniously exploiting the photochromic properties of diarylethene and the high fluorescence of rhodamine 6G. The addition of Fe^3+^ triggers a structural transition in rhodamine 6G from a non-fluorescent spirolactam to a fluorescent open form, significantly enhancing fluorescence (Figure 9). This fluorescence enhancement provides a visual indicator of Fe^3+^ presence, manifesting as a color change from colorless to pink with intense fluorescence. The sensor exhibits high selectivity and sensitivity for Fe^3+^, with a detection limit of 65 nM, and can reversibly switch between fluorescent and non-fluorescent states under UV/visible light, adding controllability and reversibility to the system. Additionally, the fluorescence response of the sensor can be modulated by other external stimuli, such as pH changes, allowing it to function dynamically in various environments. This makes it a potent tool for accurate and real-time detection of iron ions in environmental monitoring and biomedical research. Based on similar principles, this research group has also reported several other photoswitchable fluorescent probes for detecting Hg^2+^ [149,150,151], Cd^2+^ [152], and Cr^3+^ [153]. By utilizing various ligands combined with diarylethene, the Pu group has also developed additional photoswitchable fluorescent probes for detecting Hg^2+^ [154], Cd^2+^ [155,156,157], Cr^3+^ [158], and Sn^2+^ [159].

Through meticulous molecular design, combining photo-responsive azobenzene moieties with other reactive fluorescent groups, it is possible to develop photoswitchable fluorescent probes for detecting metal ions [160,161]. Raman et al. have successfully synthesized a novel molecular dyad that cleverly integrates fluorescent rhodamine with photochromic azobenzene, connected via a phenylene bridge (Figure 10) [160]. This strategic design not only maintains the unique electronic properties of each component but also creates a multifunctional platform that is responsive to a variety of stimuli, including light and Fe^3+^ ions. Experimentally, the probe showed distinct behaviors; under UV light at 365 nm, trans-azobenzene was transformed into its cis-isomer—a process that could be reversed in the absence of light. The addition of Fe^3+^ ions triggered the opening of the spirolactam ring in the rhodamine unit, leading to the formation of a 1-Fe^3+^ complex that showed significantly increased absorption at 556 nm. This complex formation effectively prevents the usual cis-trans isomerization of the azobenzene segment. Additionally, further light exposure to the 1-Fe^3+^ complex did not induce azobenzene isomerization but did result in an increased absorption at 556 nm, suggesting the formation of more open-ring amide forms of rhodamine. This observed behavior is believed to be due to energy transfer from the photoexcited azobenzene to the rhodamine, causing the rhodamine to undergo a ring-opening reaction and form a fluorescent amide variant. Remarkably, this process led to a 640-fold increase in the fluorescence intensity of rhodamine, demonstrating the efficacy of through-bond energy transfer (TBET). This pioneering study offers a robust framework for the development of advanced materials that can dynamically respond to both physical and chemical stimuli. The unique ‘turn-on’ fluorescence mechanism provided by this probe represents an effective approach for the detection of metal ions, illustrating the vast potential of integrating photochromic and fluorescent groups in sensor design.

### 3.2. Anions

#### 3.2.1. CN^−^

Cyanide ions (CN^−^) are highly toxic, inhibiting mitochondrial electron transport chains and preventing oxygen metabolism, leading to significant biological harm [162,163,164,165]. Monitoring the presence of CN^−^ is crucial for environmental safety, particularly near industrial sites like gold mines and electroplating facilities that use substantial amounts of cyanides. The World Health Organization (WHO) has established strict threshold guidelines (1.9 μM) for cyanide levels in drinking water, underscoring the necessity for accurate and reliable detection methods. In this context, photoswitchable probe molecules offer a promising method for the selective and sensitive detection of CN^−^ ions. These molecules can alter their physical properties under light exposure, which can be exploited to effectively detect and measure cyanide levels. Common photoswitchable molecules used in the construction of fluorescent probes for CN^−^ detection include spiropyran and diarylethene. The mechanisms for CN^−^ recognition involve the addition of electron-deficient conjugated systems and interaction with labile hydrogen atoms in the probe molecules that CN^−^ can bind to. Based on these mechanisms, a significant number of photoswitchable fluorescent probes for cyanide detection have been successfully developed, providing a valuable tool for environmental safety and compliance with health regulations.

In 2009, Shiraishi et al. reported a colorimetric probe based on spiropyran for the reversible, light-controlled detection of CN^−^ [166]. The probe undergoes a structural change from a colorless spiropyran (SP) form to a pink merocyanine (MC) form under UV light, which reverts back under visible light (Figure 11). The transformation to the merocyanine (MC) form allows for a nucleophilic addition reaction with cyanide ions, making the reaction visually detectable. This reaction is reversible; the probe can be regenerated by visible light irradiation without any chemical additives, enabling its reuse. The probe is capable of quantitatively determining low levels of cyanide in water, with a detection limit reported at 1.7 µM, below the safety threshold set by the World Health Organization (WHO). Overall, this study presents a viable method for the rapid, sensitive, and reversible detection of cyanide in environmental and potentially clinical settings. The use of spiropyran offers a stable and reproducible platform for further development of optical CN^−^ sensors, distinguishing it for its selective detection capabilities among potentially interfering anions. Subsequent enhancements included structural modifications to the spiropyran and the incorporation of new fluorescent groups, yielding probes with both colorimetric and fluorescent responses to cyanide [167,168].

Hu et al. developed a novel cyanine-based dithienylethene that performs photocyclization under visible light, setting it apart from traditional photochromic materials, which rely on ultraviolet light [169]. This compound exhibits dual reactivity: it can be triggered to cyclize under visible light and revert under UV light, demonstrating its multifunctional capabilities as a photochromic switch. The introduction of acceptor–π–acceptor (A–π–A) dye groups enhances the π-conjugation, improving the material’s sensitivity and allowing operation under less invasive lighting conditions. Importantly, cyanide ions can specifically engage in nucleophilic addition with the electron-deficient double bonds of the probe molecule under visible light. This interaction is distinctive, as photochromic materials typically do not selectively interact with ions during photoreactions. The reaction disrupts the extensive conjugated system of the probe molecule, significantly decreasing its UV absorption and fluorescence emission, thus enabling specific detection of cyanide ions. This study not only enhances the understanding of cyanide-responsive photochromic materials but also opens new avenues for their application in detecting harmful cyanides. Exploiting the specific nucleophilic addition of cyanide ions to distinct electron-deficient double bonds, two additional dithienylethene-based photochromic fluorescent probes for cyanide detection have been developed [170,171]. Several photoswitchable fluorescent probes were synthesized by integrating diarylethene with various structural units possessing active hydrogens, such as hydroxyl and hydrazine groups [101,172,173]. Cyanide ions interact with these active hydrogens either by forming hydrogen bonds or by abstracting hydrogen. This interaction alters the spectral properties of the probes, enabling specific responses to cyanide ions.

#### 3.2.2. F^−^

Fluoride ions (F^−^) are ubiquitous in nature, particularly in water bodies. While small amounts of fluoride can prevent dental cavities and promote oral health, excessive intake can lead to fluorosis, affecting bones and teeth and potentially causing severe diseases such as skeletal hardening and osteoporosis [174,175]. Thus, precise monitoring of fluoride ions in drinking water and the environment is crucial. According to WHO standards, fluoride levels in drinking water should be kept below 1.5 mg·L^−1^. Several photochromic fluorescence probes for detecting fluoride ions have been reported in the literature [176,177,178,179].

Zhou et al. synthesized a diarylethene compound that incorporated an organoboron group known to interact strongly with fluoride ions [176]. This interaction induces a significant change in the material’s photochromic behavior. The core of the diarylethene was modified to include boron atoms that act as strong Lewis acids, reacting distinctively with fluoride ions, which behave as Lewis bases. Upon the introduction of fluoride ions, the photochromic response of the diarylethene was modulated, which was evident from the shifting of absorption maxima in UV–visible spectrophotometry experiments. Specifically, the absorption maximum shifted from 655 nm (without fluoride) to 490 nm (with fluoride), indicating a clear modulation of the photochromic properties due to the fluoride ion interaction. The modulation mechanism was attributed to the formation of a complex between the boron atom in diarylethene and the fluoride ion. This complex formation disrupts the π-conjugation in the molecule, altering its optical properties. Guan and colleagues have developed a functional nanomaterial using spiropyran molecules and mesoporous silica powder for the specific adsorption and photo-induced desorption of fluoride ions [178]. The mesoporous silica, with a large surface area, covalently binds approximately 0.14–0.15 mmol of spiropyran per gram of silica. Under dark conditions, the spiropyran on the silica surface converts to its merocyanine form, which adsorbs fluoride ions. When exposed to visible light, this adsorption process is reversed, releasing the fluoride ions back into the water and regenerating the adsorbent. This material shows a high fluoride ion loading capacity, with successful adsorption of up to 2.3 mg·g^−1^ at an adsorbent dose of 1.0 g·L^−1^, and is capable of desorbing under visible light. Additionally, adsorption kinetics studies using empirical Freundlich and theoretical Langmuir isotherms fit best with the pseudo-first-order model.

Zhang et al. introduced an indole-hemithioindigo-based photoswitchable probe for selective detection of F^−^ and OH^−^ (Figure 12) [179]. This probe, capable of reversible E to Z isomerization in response to these ions, shows excellent selectivity and can operate reversibly across various solvents and polymer hydrogels. This unique property highlights its potential in chemical sensing, particularly for anion detection. Unlike traditional irreversible chemodosimeters and other conventional fluorescence probes, this probe’s photo-kinetic sensing relies on non-equilibrium chemical kinetics, offering a novel method to distinguish between different analytes. The probe demonstrates high stability, rapidly switching from the Z to the E isomer under visible light and slowly reverting in the dark. Its rapid and controllable response to F^−^ and OH^−^ underscores its potential for developing new biocompatible materials and environmental monitoring devices. This work showcases the potential of indole-hemithioindigo-based probes in anion sensing, particularly their ability to manipulate isomeric stability in a photo-controlled chemical environment, providing valuable insights for designing new photoswitchable chemical sensors.

#### 3.2.3. SO_3_^2^^−^ (HSO_3_^−^, SO_2_, H_2_SO_3_)

Sulfite ions (SO_3_^2−^) and their various forms (HSO_3_^−^, SO_2_, and H_2_SO_3_) are crucial in both the chemical industry and biological systems. Industrially, sulfites are extensively utilized as preservatives and antioxidants, especially in food processing and chemical manufacturing. For instance, they help prevent oxidation and maintain color stability during wine production [180]. Biologically, sulfites act as vital intermediates in sulfur metabolism. They are involved in the reduction of sulfate to sulfide, which is essential for the production of sulfur-containing biomolecules, such as methionine in proteins, within cells [181,182]. The necessity for detecting sulfites is driven by their environmental and health impacts. Environmentally, excessive sulfite release can lead to pollution of the air and water, adversely affecting both flora and fauna. From a health perspective, sensitivity to sulfites in some individuals can provoke respiratory issues, skin reactions, and other allergic symptoms, particularly in asthmatics. Therefore, accurate detection of sulfites in food, environmental samples, and biological matrices is essential for public health and environmental monitoring [183,184,185]. Furthermore, the development of effective in-situ analytical methods for sulfites is crucial for a deeper understanding of their biological roles [186,187]. Photochromic molecular probes offer a dynamic and flexible approach to the detection of sulfite and bisulfite ions. These probes allow for real-time monitoring of sulfite levels; adjustments in light exposure can instantly alter the probe’s active state, enabling it to respond adaptively to changes in sulfite concentrations in the environment. Spiropyrans, in particular, are excellent candidates for crafting such probes. They can be optically manipulated to switch between their spiropyran (SP) and merocyanine (MC) forms, with the MC form being particularly reactive towards sulfite ions. Recent developments have led to the creation of several spiropyran-based photochromic probes for sulfites, which have been successfully used for in-situ biological imaging and analysis.

Zhang et al. have presented an innovative approach to enhance the accuracy and reliability of intracellular imaging and biosensing through the dynamic and reversible alteration of probe properties using UV and visible light [188]. The strategy centers around a photochromic glycoprobe (SP-Gal) composed of a naphthalimide fluorophore linked to a spiropyran, with a glycoligand addition for targeted cellular uptake. Key to the probe’s function is its ability to switch between two isomeric forms—spiropyran (SP) and merocyanine (MC)—under light control (Figure 13). In the absence of UV light, the probe remains in the inactive SP state and does not interact with the target sulfite ion. UV light exposure triggers a change in the MC state, enabling specific interactions with sulfite ions. This switchable “on” and “off” fluorescence capability, driven by light rather than environmental factors, allows for precise cellular imaging and analysis while avoiding the typical pitfalls of environmental interference. This dual-state functionality is utilized to perform targeted imaging within cells that express specific glycoligand receptors, enhancing the selectivity and effectiveness of the imaging process. The glycoprobe’s fluorescence can be toggled by the wavelength of light applied, offering a method to remotely control and identify intracellular sulfite ions, a capability that could significantly improve understanding in various biomedical research fields. This development represents a significant advancement in bioimaging, providing researchers with a powerful tool for detailed cellular studies. Following the initial development, this research group discovered that the glycoprobe SP-Gal effectively binds with human serum albumin (HSA), forming a glycoprobe–protein hybrid [189]. This hybrid utilizes a unique “double-check” mechanism to achieve reversible fluorescence switching under UV and visible light, significantly enhancing the accuracy and reliability of biological imaging. The design of this probe allows it to specifically target cells expressing glycoligand receptors and perform precise imaging and tracking within these cells. These studies not only demonstrate the potential applications of photo-controlled fluorescence probes in biomedical imaging but also provide a crucial technological foundation for the future development of advanced imaging systems that can dynamically respond to biochemical signals. Tang et al. have reported a new fluorescent probe system (HSP) that combines naphthalimide fluorescence interactions with HSA, fluorescence resonance energy transfer effects between naphthalimide and spiropyran, and the light-controlled fluorescence characteristics of spiropyran for selective and controllable detection of SO_2_ [190]. When HSP interacts with HSA, there is a notable increase in fluorescence at 540 nm. This fluorescence can be further controlled under UV light, which transforms the spiropyran into its merocyanine form, altering the probe’s fluorescence characteristics (emission at 610 nm). This double fluorescence transition (initially at 540 nm and then shifting to 610 nm under UV light) underpins the “double-check” mechanism, which ensures more precise and reliable HSA detection. Additionally, the HSP probe is employed for real-time cellular imaging of SO_2_, showcasing its versatility and dual functionality. The probe responds to SO_2_ presence by altering its fluorescence, enabling real-time monitoring and providing a valuable tool for cellular studies involving SO_2_.

Researchers have developed additional effective ratiometric fluorescence detection systems for sulfites by combining spiropyran with other fluorophores and incorporating them into nanomaterial platforms. Xu and Zhang have developed a novel nanoprobe, Poly-Cm-SP, for detecting and imaging intracellular sulfites [191]. Initially, they synthesized the photochromic molecule Cm-SP by linking a coumarin fluorophore to spiropyran, which was subsequently co-precipitated with the polymer mPEG-b-PBMA to form the Poly-Cm-SP nanoprobe. This probe can switch between red and blue fluorescence under UV and visible light irradiation, a process reliant on Förster Resonance Energy Transfer (FRET) between coumarin and the spiropyran/merocyanine (SP/MR) isomers. The presence of sulfite ions reacts with the electron-deficient ‘C=C’ bond in the MR isomer, disrupting the FRET process and maintaining stable blue fluorescence, which indicates the presence of sulfites. This mechanism allows for ratiometric fluorescence detection by monitoring changes in the intensity of red and blue light to determine sulfite levels. The Poly-Cm-SP nanoprobe exhibits excellent biocompatibility and is localized within the lysosomes of macrophages, which is crucial for its application in live-cell imaging. These probes can image intracellular sulfites, providing valuable insights into cellular dynamics and the biochemical environment related to sulfite metabolism. Li et al. innovated on the traditional spiropyran molecule by replacing its central cyanine part with a larger, more rigid azahomoadamantane structure, thus developing a new type of negative photochromic spiropyran derivative named spiro[azahomoadamantane-pyran] (MAHD-SP) [192,193]. They combined MAHD-SP with the fluorescent polymer PFO and the surfactant polymer PEO113-b-PS49 using a co-precipitation method to create nanoprobes (RFPN) for ratiometric detection of sulfite ions [193]. In this nanoprobe, PFO acts as the energy donor and effectively transfers energy to the stable open-ring form of MAHD-SP through Förster Resonance Energy Transfer (FRET), resulting in orange fluorescence. When SO_2_ derivatives are present, they react with MAHD-SP via a Michael addition, disrupting its conjugated structure and extinguishing the fluorescence, thereby halting the FRET process and causing the probe to emit blue fluorescence. This mechanism allows the probe to perform ratiometric fluorescence detection of sulfite ions. Additionally, the excellent biocompatibility of RFPN enables it to image exogenous and endogenous SO_2_ in living cells in real-time, providing a powerful tool for biomedical research.

#### 3.2.4. S^2^^−^ (HS^−^, H_2_S)

Sulfide ions (S^2^^−^) and their various forms (H_2_S, HS^−^) play crucial roles in both natural environments and biological systems. Hydrogen sulfide is not only a pervasive small molecule involved in numerous biological processes such as neurotransmission, inflammation, and cellular protection but also serves significant industrial applications in fields like metal processing and petroleum refining. As a metabolic product of cysteine and methionine, H_2_S can regulate the activity of sulfide enzymes and cell apoptosis. Hydrosulfide, the partially dissociated form of hydrogen sulfide in water, is closely linked to various biochemical processes through its concentration changes. While hydrogen sulfide is a vital biological signaling molecule, its high concentrations can cause serious environmental and health issues, such as severe respiratory irritation and poisoning, making it essential to monitor its levels in the environment for protection and safety. Accurate detection of H_2_S is also crucial for understanding its roles in regulating blood pressure, heart, and brain functions [194,195,196]. However, reports on the use of photoswitchable probes in detecting hydrogen sulfide are still relatively scarce. Hong et al. have developed a new type of photoswitchable aggregation-induced emission (AIE) nanoprobe, DNBS-DCM-SP, specifically designed for high-specificity and spatiotemporal resolution detection of hydrogen sulfide (H_2_S) in lysosomes [197]. This nanoprobe combines an H_2_S-sensitive AIE luminophore with a photochromic spiropyran unit, enabling it to display reversible dual-color fluorescence under UV and visible light control (Figure 14a). The probe significantly responds to H_2_S at 592 nm due to a cleavage reaction induced by H_2_S, which removes the electron-withdrawing DNBS group, facilitating precise in vivo tracking of H_2_S and enhancing understanding of its biological roles. Moreover, this probe design overcomes common fluorescence probe issues like aggregation-caused quenching (ACQ) and background fluorescence interference. Zhang et al. developed the photo-controlled fluorescent probe NT-N3-SP for detecting hydrogen sulfide (H_2_S) and sulfur dioxide (SO_2_) using a unique light-switching mechanism (Figure 14b) [198]. By integrating 4-azophenyl-1,8-naphthalic anhydride and spiropyran, the probe can reversibly alter its fluorescence under alternating UV and visible light. Exposure to H_2_S triggers a reduction in the azide group within NT-N3-SP, significantly enhancing fluorescence at 540 nm—a direct indicator of H_2_S presence that can be quantitatively measured. The probe’s rapid response is crucial for live-cell imaging and in vivo applications. Additionally, the probe’s spiropyran component reacts with SO_2_, providing specificity and enabling distinct biological role analysis of H_2_S and SO_2_.

### 3.3. Small Molecules

#### 3.3.1. Amines

Amines, such as biogenic amines and numerous medicinal molecules, are essential components of biochemical processes. They act as neurotransmitters, hormones, and signaling molecules between cells in living organisms [199,200,201]. Amines are also extensively used in industries, for example in the production of dyes, plastics, and pesticides. Detecting amines is crucial for environmental monitoring, food safety, and health care [202]. Photoswitchable molecules also demonstrate the potential for detecting amines. In 2018, Fredrich et al. developed sensitive assays based on nucleophile-induced rearrangement of photoactivated diarylethenes (DAEs) [203]. These compounds are specifically optimized for detecting amines and thiols via a light-induced mechanism. The research involved synthesizing and using diarylethenes that change structure when exposed to amines, facilitated by light activation. This structural change often results in a visible color change, thereby enhancing detectability. The researchers optimized the chemical structure of these probes to improve their inherent reactivity and selectivity for nucleophile-triggered reactions. They experimented with various aryl groups and the positioning of the formyl group within the molecule to refine the probe’s reactivity with primary and secondary amines. An innovative aspect of their methodology includes a secondary catalytic cycle that uses the initial rearrangement product to further trigger reactions, amplifying the method’s sensitivity (Figure 15). This allows for a better distinction between amines and thiols. The reaction mechanism involves the photoinduced isomerization of DAEs, followed by rearrangement in the presence of amines to produce a less colored or colorless form. This process is enhanced by using an auxiliary base to help catalytically recycle the active form of the probe. This research significantly contributes to the field of chemical sensing by providing a method that combines photochemical activation with specific molecular rearrangements to achieve highly sensitive and selective detection of biologically and industrially relevant chemicals.

Donor–Acceptor Stenhouse Adducts (DASAs) are significant photochromic molecules composed of a donor amine and an acceptor carbonyl connected by a triene-phenol skeleton [78]. These molecules can be formed by the reaction of an electron-deficient furan base with amines. Thus, utilizing the formation reaction of DASAs with furan as the recognition substrate allows for the detection of amines [204]. Cai et al. investigated the development and utilization of novel activated furan copolymers as colorimetric sensors for amines [117]. These copolymers utilize Donor–Acceptor Stenhouse Adducts (DASAs) formed by a reaction between an electron-deficient furan and an amine molecule. The formed DASAs, consisting of a donor amine linked to a carbonyl acceptor through a triene-phenol backbone, enable the detection of amines using furan as the recognition element. The copolymers, grafted with furan units, react with amines in aqueous solutions to create highly colored DASA complexes that respond to changes in pH, temperature, or light by altering their color, which is visible to the naked eye. The polymer structure is designed to enhance stability and prevent dye loss. Furthermore, the synthesis process of these polymers and their application in detecting amine compounds are elaborately described, including their use in films that act as sensors by changing color upon amine contact. These advancements provide practical tools for chemical detection in the environment. Additionally, using the mechanism of Donor–Acceptor Stenhouse Adduct formation, methods were developed for detecting mesalazine [205] and amphetamine-type stimulant (ATS) amines [206].

#### 3.3.2. Thiols

Thiol small molecules, including biothiols such as cysteine, homocysteine, and glutathione, as well as industrial thiophenols, are crucial for both biological systems and industrial applications [207,208,209,210,211,212]. Inside living organisms, these thiol compounds mainly act as antioxidants to maintain cellular redox balance and participate in various biochemical reactions; accurate detection of these compounds can help study cellular responses to disease and stress. Industrially, thiophenol is commonly used in the production of dyes, rubber, pharmaceuticals, and pesticides. It is also used to produce preservatives and antioxidants to improve product stability and shelf life. Given their high reactivity and potential toxicity, industrial emissions of thiophenols can contaminate water bodies and the atmosphere, making it essential to monitor their concentrations for environmental protection and health safety [213,214,215]. The development of photoswitchable probes has provided effective tools for detecting these small thiol molecules [216,217]. 

Zhai et al. developed a novel probe based on a bisthienylethene compound (D-HBT-NBD), innovatively combining a diarylethene unit with an NBD (nitrobenzofurazan) moiety, which is typically used to differentiate between GSH, Cys, and Hcy (Figure 16) [218]. The D-HBT-NBD probe is capable of simultaneously detecting and distinguishing three key biothiols: cysteine (Cys), homocysteine (Hcy), and glutathione (GSH). This probe exhibits remarkable photochromic properties, allowing direct observation of distinct color changes among Cys, Hcy, and GSH in its ring-open isomeric 1O state with the naked eye. Under UV light, the probe’s closed-ring state transitions from a light orange to a deep pink, effectively differentiating Cys from Hcy/GSH. Moreover, at an excitation wavelength of 465 nm, the probe distinctly responds to Cys and Hcy in fluorescence, separating GSH from Cys/Hcy. These characteristics make the D-HBT-NBD probe a powerful tool for potential applications in biological labeling and clinical diagnostics. Additionally, the study introduces a logic gate based on the 1O probe that utilizes the probe’s color and fluorescence changes to perform logical operations, paving the way for the development of innovative biosensors.

Fu et al. reported a novel photochromic probe, Np-SP-TP, specifically designed for the ratiometric and ‘double-check’ detection of thiophenol derivatives [219]. The probe is engineered by linking a naphthalimide fluorophore with a spiropyran unit, and a target recognition group, 2,4-dinitrobenzenesulfonyl (DNBS), is attached to the naphthalimide part. This design allows the Np-SP-TP system to exhibit a dual fluorescence signal mechanism that is activated only upon interaction with target phenylthiol derivatives, enabling dynamic and reversible ‘double-check’ capabilities that significantly reduce false positives. The operation of the probe involves the cleavage of the probe molecule in the presence of thiophenol, followed by light-induced isomerization between the spiropyran and merocyanine (SP/MR) forms, allowing the fluorescence emission color to switch (green to red) depending on the presence of the target thiol. The probe demonstrates high sensitivity (the detection limit for thiophenol is 6.1 nM) and excellent selectivity, effectively differentiating other common thiols such as GSH, Cys, and Hcy. It has been successfully tested in real water samples and live cells, proving its practical application capabilities for detecting trace amounts of harmful thiophenol derivatives. Furthermore, its use in confocal laser scanning microscopy and flow cytometry validates its effectiveness in live cell imaging and precise thiol detection. The probe system introduces an ‘analyte-activated’ mode, allowing the probe’s fluorescence to be activated only in the presence of specific thiols, enhancing the accuracy and reliability of detection. The ratiometric approach helps to eliminate variations caused by environmental changes, thereby stabilizing the detection process across different conditions. Following this, Guo et al. reported on a photoluminescent probe based on a cyclometalated iridium(III) complex specifically designed for selective detection of thiophenol [220]. The probe molecule incorporates two photochromic diarylethene (DTE) units and a 2,4-dinitrobenzenesulfonyl (DNBS) group. The electron-withdrawing effect of the DNBS group inhibits fluorescence due to an internal charge transfer (ICT) mechanism. Interaction with thiophenol disrupts this ICT process by eliminating the DNBS group, leading to enhanced fluorescence centered at 450 nm. The probe can switch its fluorescence properties under UV and visible light, attributed to the photochromic DTE units in its structure, allowing for tunable fluorescence behavior during detection. This capability enhances the multifunctionality of the probe. The designed probe system demonstrates a rapid response time (about 15 min) and high sensitivity (detection limit: 2.43 µM) towards thiophenol, with excellent selectivity over other types of thiols such as aliphatic thiols, which typically do not trigger the fluorescence mechanism used by this probe. The study underscores the potential applications of the probe in environmental monitoring and safety, as it can efficiently detect the toxic pollutant thiophenol.

#### 3.3.3. Diethyl Cyanophosphonate

Balamurugan et al. have developed a novel polymeric probe (P3) based on a visible light-responsive Donor–Acceptor Stenhouse Adduct (DASA) (Figure 17) [221]. This probe can rapidly and selectively detect nerve agent simulants in both liquid and vapor phases, extending its practical applications in safety and environmental monitoring. Initially, through reversible addition–fragmentation chain transfer (RAFT) polymerization, a copolymer was synthesized from glycidyl methacrylate (GMA) and dimethylacrylamide (DMA). This copolymer was then chemically modified to incorporate DASA units that are sensitive to nerve agent simulants like diethyl cyanophosphonate (DCNP). The principal function of this probe is its DASA unit, which changes color from purple to colorless when exposed to DCNP. This color change is due to an intramolecular N-alkylation reaction induced by DCNP, creating a morphenium ion. Under visible light, this polymer demonstrates excellent photochromic properties, with the DASA units undergoing transitions between different chemical states—colored and colorless. However, only the colored state reacts to the target. This capability of switchable colorimetric detection offers high sensitivity and selectivity in detecting DCNP in both liquid and vapor forms, with little to no response to similar agents. The paper discusses the potential for developing new types of switchable polymeric photoswitches for colorimetric detection of nerve agent simulants using this polymer probe. This research significantly advances the field of chemical warfare agent detection by introducing a new material that combines stability and responsive behavior in polymers. This probe’s ability to detect nerve agents in multiple states positions it as a promising candidate for real-world applications where rapid and reliable detection of hazardous substances is crucial.

### 3.4. Biomacromolecules

#### 3.4.1. β-Galactosidase

β-Galactosidase (β-Gal) is an essential enzyme that catalyzes the hydrolysis of β-galactosides into monosaccharides, playing a pivotal role in the metabolism of lactose in various organisms, including bacteria and humans. In medical and biological research, β-Gal serves as a crucial biomarker for monitoring cellular processes such as senescence, differentiation, and infection status. Its activity is commonly exploited in senescence-associated β-galactosidase staining, which is a hallmark of cellular aging and is used extensively to identify senescent cells in culture and in tissue samples. The detection of β-Gal activity is not only important for understanding basic biological processes but also has implications for diagnosing and tracking diseases, particularly those related to aging like neurodegenerative diseases, cancer, and conditions leading to tissue degeneration [222,223,224]. Chai et al. developed a novel photochromic fluorescent probe named NpG, specifically engineered for targeting and visualizing β-Gal activity at super-resolution levels within biological samples [225]. The probe consists of a naphthalimide fluorescence group and a spiropyran unit, with a galactose moiety linked to the phenolic hydroxyl group of the open form of spiropyran, specifically targeting β-Gal (Figure 18a). The NpG probe can form a complex with human serum albumin (HSA), resulting in the NpG@HSA probe/protein complex (Figure 18b). The formation of this complex enhances fluorescence emission at 520 nm, corresponding to the binding of the fluorescent naphthalimide unit with HSA. This probe/protein complex approach creates a unique imaging platform with enhanced cellular permeability and solubility, enabling observation of NpG@HSA uptake by cells prior to β-Gal activation. Upon β-Gal-mediated cleavage of the galactose unit, the NpG@HSA complex transforms into NpM@HSA, which shifts the fluorescence emission to 620 nm, indicating the merocyanine form of spiropyran necessary for light-induced switching in super-resolution imaging. The probe exhibits dual emission properties, where the presence of β-Gal triggers a switch from green to red fluorescence, allowing precise imaging of enzyme activity. The unique design of the probe combines photochromic properties with enzymatic reactivity, enabling it to perform imaging beyond the diffraction limit using Stochastic Optical Reconstruction Microscopy (STORM). The probe demonstrates high photostability and low phototoxicity during imaging, making it suitable for prolonged studies in living cells. The performance of the probe across various cell lines has been validated, showing its capability to detect and image β-Gal at subcellular distribution with nanoscale precision. This ability provides significant insights into cellular functions and disease states, highlighting the probe’s potential in biomedical research and diagnostic applications. This research significantly contributes to the field by merging molecular engineering with advanced optical techniques, offering highly sensitive and selective detection of biologically relevant targets.

#### 3.4.2. β-Amyloid Protein

The Aβ protein is a series of peptide fragments produced by the enzymatic cleavage of amyloid precursor protein by β-secretase and γ-secretase, primarily forming the core component of the amyloid plaques in the brain [226,227]. During the pathogenesis of Alzheimer’s disease (AD), abnormal aggregation of Aβ leads to the formation of neurofibrillary tangles and plaques, which are pathological hallmarks of the disease. Detecting Aβ is crucial for the early diagnosis and monitoring of AD progression. It also enables researchers to better understand the pathophysiological mechanisms of AD, which aids in developing more effective treatments. Traditional imaging techniques such as PET and SPECT, although capable of detecting Aβ, are costly and complex. In contrast, fluorescence imaging techniques are emerging as powerful tools for studying Aβ aggregation due to their lower cost, ease of use, and lack of radioactive exposure. Fluorescent probes, in particular, can specifically label Aβ, making them highly useful in super-resolution microscopy, offering more detailed pathological insights at the cellular level than traditional imaging techniques [228,229].

Lv et al. developed two novel fluorescent switchable probes, T1 and T2, by integrating aminonaphthalenyl-2-cyano-acrylate (ANCA) units, which target amyloid-beta (Aβ) deposits, with diarylethene (DAE) molecules known for their photochromic properties (Figure 19) [230]. These probes exhibit high affinity towards Aβ aggregates, demonstrated by significant increases in fluorescence intensity and slight shifts towards blue wavelengths upon binding (Figure 19b–d). They also displayed exceptional photochromic and anti-photobleaching abilities both in vitro and within brain tissue. The DAE component allows the probes to undergo reversible photo-induced cyclization and cycloreversion under alternating ultraviolet and visible light. This transition toggles the molecular structure between open-ring (high fluorescence) and closed-ring (low fluorescence) states, enabling prolonged and real-time bioimaging (Figure 19e). These switchable probes offer a clear advantage over traditional always-on probes by minimizing background interference and are potentially suitable for super-resolution imaging applications. Furthermore, the probes’ interaction with Aβ aggregates was analyzed using fluorescence spectroscopy, which revealed that binding significantly increased fluorescence intensity and induced a subtle blue shift in emission peaks. The probes demonstrated a strong affinity for Aβ aggregates, facilitated by the synergistic binding effects of DAE and ANCA to the hydrophobic pockets of the aggregates, leading to enhanced fluorescence. Unlike other proteins, such as prions, the probes showed high specificity for Aβ, suggesting their utility in tracing the aggregation process of Aβ with minimal interference. This development marks a significant step forward in the specific detection of Aβ aggregates in Alzheimer’s disease, enhancing our understanding of its pathophysiology and providing new tools for diagnosis and research.

Dreos et al. developed a novel photoswitchable fluorescent norbornadiene derivative (NBD1), designed as a multifunctional probe targeting the characteristic amyloid-β (Aβ) plaques of Alzheimer’s disease pathology (Figure 20) [231]. This probe exhibits highly microenvironment-sensitive fluorescence, with emission properties that vary according to the surrounding environment. Utilizing this characteristic, researchers were able to depict the structural heterogeneity of Aβ plaques in Alzheimer’s disease mouse models. NBD1 was tested on brain tissue sections from transgenic Alzheimer’s disease mouse models, successfully imaging the amyloid plaques. The spectral variations in the probe revealed significant differences between the plaque peripheries and dense cores, indicating interactions with various microenvironments. Additionally, NBD1 demonstrated in situ photoisomerization and thermal recovery within the plaques, which is crucial for applications requiring high resolution and dynamic imaging capabilities. Molecular docking studies further validated the binding mechanism of NBD1 with amyloid plaques, revealing its interactions with both hydrophobic and polar residues within the plaques, thus enhancing its photophysical properties and targeting specificity. The unique properties of NBD1 make it suitable for super-resolution microscopy and other advanced imaging techniques, potentially providing new insights into the heterogeneity and molecular changes in amyloid plaques. This research paves the way for using NBD1 and similar photoswitchable fluorescent probes for high-resolution imaging of amyloid plaques, as well as the diagnosis and study of Alzheimer’s disease.

#### 3.4.3. Nucleic Acids

Nucleic acids, such as DNA and RNA, are critical biomolecules in all forms of life, carrying genetic information and participating in various cellular processes, including protein synthesis, gene regulation, and acting as catalysts in biochemical reactions. Utilizing external stimuli like light to precisely control the function and structure of nucleic acids is an active area of research in the biosciences. Photoswitchable molecules, which change their chemical structure under specific wavelengths of light, provide a powerful tool for controlling nucleic acid functions. These molecules often contain functional groups such as azobenzene, capable of switching between trans and cis configurations under UV or visible light. Integrating these photoswitchable molecules into nucleic acids or designing them to bind specifically to nucleic acids allows for the reversible control of nucleic acid structure and function [232,233,234,235]. For example, light can be used to modulate the binding affinity of a photoswitchable molecule to a nucleic acid, thereby regulating gene expression, protein synthesis, or the structural conformation of nucleic acids. The application of photoswitchable molecules in constructing nucleic acid sensors holds significant potential. These sensors can detect specific sequences of DNA or RNA by altering their fluorescence or other optical properties upon nucleic acid binding. The integration of photoswitchable elements enables these sensors to operate in a “reversible” manner, where the sensor’s active state can be controlled by light, which is especially beneficial for reusable diagnostics and applications requiring continuous monitoring.

Cui et al. introduced an enhanced method for single-cell transcriptomics using fluorescence in situ hybridization (FISH), termed fliFISH [236]. This approach leverages photoswitchable dyes and temporal fluctuations in fluorescence emissions to accurately distinguish true RNA signals from background noise, thus improving the precision and reliability of counting RNA copies within individual cells. Photoswitchable dyes are attached to FISH probes in this method, which can be repeatedly switched on and off, generating a quantifiable blinking pattern. Emphasis is placed on the on-time fractions of these blinking dyes, effectively differentiating them from the background fluorescence. This is particularly important as it addresses the challenges posed by nonspecific probe binding and autofluorescence, which often complicate traditional smFISH. By precisely measuring actual RNA signals against background signals within a diffraction-limited area with minimal probe requirements, the technique has proven valuable. Applied to examine gene expression patterns in mouse pancreatic islets, the technique has revealed radial gene expression patterns of insulin and transcription factors such as NKX2-2, which remained indiscernible using conventional methods in diabetic mouse models. Through its significant enhancement of the reliability of RNA quantification in single cells, especially under conditions of high background fluorescence or when detecting low-copy transcripts, fliFISH represents a promising tool in medical diagnostics and cellular biology. The method provides a robust platform for elucidating cellular heterogeneity and the dynamics of disease-related gene expression at the single-cell level, showcasing the potential applications of utilizing photoswitchable dyes in biomedical research.

Guo et al. developed a novel method for high-contrast detection of nucleic acids in complex biological samples, using photoswitch-mediated FRET to address common issues of autofluorescence and light scattering in traditional fluorescence detection techniques [237]. This method centers around the use of photoswitchable nanoprobes composed of a photochromic naphthopyran molecule integrated with a brush copolymer, polystyrene-graft-poly(ethylene oxide). In this setup, the photochromic molecule acts as a fluorescence quencher in its open form under UV light, enabling a wash-free, high-contrast detection mechanism that robustly handles background fluorescence. The detection mechanism exploits FRET between the open form of naphthopyran (the photoswitch) and fluorescent groups on the nanoprobe surface. This design allows for selective optical signal retrieval even under strong background interference, thereby enhancing the signal-to-noise ratio. This method demonstrated improved sensitivity and a reduced detection limit compared to conventional FRET-based methods. It was successfully evaluated in both sandwich hybridization assays and label-free DNA detection using a nucleic stain. The nanoprobes can be prepared and interrogated in capillaries, indicating potential for high-throughput applications. Experiments included measuring fluorescence changes upon UV irradiation, showcasing the capability to accurately quantify nucleic acids amidst interfering backgrounds, including tests in complex sample matrices like blood serum, proving the method’s practical applicability in real-world scenarios. This research advances the field of nucleic acid detection by introducing a novel photoswitching technique, offering a powerful tool for clinical diagnostics and gene therapy.

## 4. Conclusions and Perspectives

This review systematically presents the applications and advancements of photoswitchable fluorescent and colorimetric probes in the detection and imaging of biological and environmental targets. These probes, capable of toggling between different fluorescent states under light control, represent significant progress in molecular sensing and imaging techniques. Their unique properties, such as high sensitivity, selectivity, and real-time monitoring capabilities, make them indispensable tools in medical diagnostics, environmental monitoring, and biochemical research. The key feature of these probes is their ability to be reversibly activated by light, allowing for dynamic studies with minimal disturbance to biological systems. This attribute is particularly advantageous in super-resolution microscopy and long-term imaging studies, where phototoxicity and photobleaching are major concerns. Additionally, the versatility of these probes is demonstrated by their adaptability to a wide range of targets, from ions and small molecules to proteins and other macromolecules, highlighting their broad application potential.

Despite considerable progress in the study of photoswitchable probes, there are still challenges, such as the need for organic solvents to better demonstrate the photoswitching properties and the restriction of controlling light wavelengths to the shorter UV and visible spectra, limiting broader applications. Future research could be directed in the following areas:

(1) Further exploration of the photophysical properties and principles of photoswitchable molecules is needed to understand the effects of solvent cutoff and develop high-performance photoswitchable molecular probes that are more adaptable, particularly to aqueous and biological systems.

(2) Modification of photoswitchable molecules to adjust the control light wavelengths, allowing these molecules to respond to longer wavelengths of visible light or even near-infrared light, which is crucial for applications in complex environmental and biological samples.

(3) Integration of photoswitchable molecular probes with nanomaterials, including polymers and inorganic nanomaterials, to enhance the stability and environmental suitability of the probes. Combining photoswitchable molecular probes with upconversion nanomaterials to achieve indirect control by near-infrared light is also a viable strategy.

(4) Expansion of the types of photoswitchable fluorescent probes used, moving beyond commonly employed molecules like diarylethenes and spiropyrans to explore new types of molecular switches such as hemithioindigos, norbornadienes, and furylfulgimides.

(5) Utilization of various energy transfer processes, such as fluorescence resonance energy transfer (FRET), through-bond energy transfer (TBET), and ligand–metal charge transfer (LMCT), combined with other high-performance fluorophores to construct a richer array of photoswitchable fluorescent probes.

(6) Combination of photoswitchable molecules with other types of luminescence, such as chemiluminescence and electrochemiluminescence, to establish a more diverse set of photo-controlled detection systems.

## Data Availability

Not applicable.

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
