# Peer review of "Recent Advances in Photoswitchable Fluorescent and Colorimetric Probes"

_molecules, 2024, doi:10.3390/molecules29112521_

Round 1
Reviewer 1 Report
Comments and Suggestions for Authors
Overall, this manuscript focuses on photoswitchable fluorescent and colorimetric probes and provides a clear introduction about recent advances in the field. The review not only introduces the types/construction of photoswitchable probes, but also discusses the applications for sensing and imaging. The structure of this review article is very well organized, and the content is informative, which will be friendly to readers. Thus, this work is worth of publication in Molecules. However, the following minor concerns should be addressed before publication.
1. The concept of photoswitchable probe is great for sure. However, most of the probes are triggered by short wavelength light source (especially UV), which is not biofriendly. Could the author introduce some cases that use longer wavelength light? What are the challenges to develop such probes?
2. In section 3.3, the authors introduce some applications about protein sensing. Is there any case about photoswitchable probes for DNA or RNA sensing or imaging? Since it can be significant to observe/control gene expression (transcription and translation), and this will broaden the research interests in the field. If not, please add some comments in the conclusion part.
3. Minor issue about the section numbering. After 3.2 or 3.3, there should be 3.2.1 or 3.3.1. Now, they are all 3.1.x. Please revise.
4. Please check the figure captions. If a figure is from literature, please add references.
Comments on the Quality of English LanguageOverall, the language is OK. Only minor editing is needed.
Author Response
Reviewers’ comments
Reviewer 1
Overall, this manuscript focuses on photoswitchable fluorescent and colorimetric probes and provides a clear introduction about recent advances in the field. The review not only introduces the types/construction of photoswitchable probes, but also discusses the applications for sensing and imaging. The structure of this review article is very well organized, and the content is informative, which will be friendly to readers. Thus, this work is worth of publication in Molecules. However, the following minor concerns should be addressed before publication.
Comment 1: The concept of photoswitchable probe is great for sure. However, most of the probes are triggered by short wavelength light source (especially UV), which is not biofriendly. Could the author introduce some cases that use longer wavelength light? What are the challenges to develop such probes?
Response: Thank you very much for your professional and constructive comments, and for your appreciation and recognition of our paper. Our review primarily focuses on photocontrollable fluorescent and colorimetric probes that can selectively respond to specific targets, incorporating photocontrol in the detection and/or signal output processes. Currently, these probes largely utilize classical photochromic molecules that respond to UV light, which indeed presents certain limitations for biological applications. To address this issue, our review specifically highlights the structural modification and transformation of classical photochromic molecules to respond to visible and even near-infrared light. Although improvements in photochromic molecules have been reported, developments in constructing photocontrollable fluorescent and colorimetric probes are still rare. Following your valuable suggestions, we have further emphasized discussions on long-wavelength responsive photochromic molecules and photocontrollable probes in both the introduction and the conclusions and future outlook sections of our paper.
“Recent studies have shown that by modifying the structure and adjusting the conjugation, the responsive wavelength of photoswitches can be extended from the ultraviolet to the visible and even near-infrared spectrum [40-42]. This extension provides a crucial basis for deeper tissue photocontrol imaging and detection.”
“(2) Modification of photoswitchable molecules to adjust the control light wavelengths, allowing these molecules to respond to longer wavelengths of visible light or even near-infrared light, which is crucial for applications in complex environmental and bio-logical samples.”
Comment 2: In section 3.3, the authors introduce some applications about protein sensing. Is there any case about photoswitchable probes for DNA or RNA sensing or imaging? Since it can be significant to observe/control gene expression (transcription and translation), and this will broaden the research interests in the field. If not, please add some comments in the conclusion part.
Response: Thank you for your insightful comment. Existing research has demonstrated the significant role of photoswitchable molecules in regulating the structure and function of nucleic acids. In our initial literature review, we did not find any direct reports of photoswitchable probes being used for DNA or RNA detection. However, upon your suggestion, we have revisited the literature and discovered various sensor systems utilizing photoswitchable molecules for nucleic acid detection. We have included examples of these in the revised manuscript.
“3.3.3. Nucleic acids
Nucleic acids, such as DNA and RNA, are critical biomolecules in all forms of life, carrying genetic information and participating in various cellular processes, including protein synthesis, gene regulation, and acting as catalysts in biochemical reactions. Uti-lizing external stimuli like light to precisely control the function and structure of nucleic acids is an active area of research in the biosciences. Photoswitchable molecules, which change their chemical structure under specific wavelengths of light, provide a powerful tool for controlling nucleic acid functions. These molecules often contain functional groups such as azobenzene, capable of switching between trans and cis configurations under UV or visible light. Integrating these photoswitchable molecules into nucleic acids or designing them to bind specifically to nucleic acids allows for the reversible control of nucleic acid structure and function [232-235]. For example, light can be used to modulate the binding affinity of a photoswitchable molecule to a nucleic acid, thereby regulating gene expression, protein synthesis, or the structural conformation of nucleic acids. The application of photoswitchable molecules in constructing nucleic acid sensors holds sig-nificant potential. These sensors can detect specific sequences of DNA or RNA by altering their fluorescence or other optical properties upon nucleic acid binding. The integration of photoswitchable elements enables these sensors to operate in a "reversible" manner, where the sensor's active state can be controlled by light, which is especially beneficial for reusa-ble diagnostics and applications requiring continuous monitoring.
Cui et al. introduced an enhanced method for single-cell transcriptomics using flu-orescence in situ hybridization (FISH), termed fliFISH [236]. This approach leverages photoswitchable dyes and temporal fluctuations in fluorescence emissions to accurately distinguish true RNA signals from background noise, thus improving the precision and reliability of counting RNA copies within individual cells. Photoswitchable dyes are at-tached to FISH probes in this method, which can be repeatedly switched on and off, gen-erating a quantifiable blinking pattern. Emphasis is placed on the on-time fractions of these blinking dyes, effectively differentiating them from the background fluorescence. This is particularly important as it addresses the challenges posed by nonspecific probe binding and autofluorescence, which often complicate traditional smFISH. By precisely measuring actual RNA signals against background signals within a diffraction-limited area, with minimal probe requirements, the technique has proven valuable. Applied to examine gene expression patterns in mouse pancreatic islets, the technique has revealed radial gene expression patterns of insulin and transcription factors such as NKX2-2, which remained indiscernible using conventional methods in diabetic mouse models. Through its significant enhancement of the reliability of RNA quantification in single cells, especially under conditions of high background fluorescence or when detecting low-copy transcripts, fliFISH represents a promising tool in medical diagnostics and cellular biolo-gy. The method provides a robust platform for elucidating cellular heterogeneity and the dynamics of disease-related gene expression at the single-cell level, showcasing the poten-tial applications of utilizing photoswitchable dyes in biomedical research.
Guo et al. developed a novel method for high-contrast detection of nucleic acids in complex biological samples, using photoswitch-mediated FRET to address common is-sues of autofluorescence and light scattering in traditional fluorescence detection tech-niques [237]. This method centers around the use of photoswitchable nanoprobes com-posed of a photochromic naphthopyran molecule integrated with a brush copolymer, polystyrene-graft-poly(ethylene oxide). In this setup, the photochromic molecule acts as a fluorescence quencher in its open form under UV light, enabling a wash-free, high-contrast detection mechanism that robustly handles background fluorescence. The detection mechanism exploits FRET between the open form of naphthopyran (the pho-toswitch) and fluorescent groups on the nanoprobe surface. This design allows for selec-tive optical signal retrieval even under strong background interference, thereby enhancing the signal-to-noise ratio. This method demonstrated improved sensitivity and a reduced detection limit compared to conventional FRET-based methods. It was successfully evalu-ated in both sandwich hybridization assays and label-free DNA detection using a nucleic stain. The nanoprobes can be prepared and interrogated in capillaries, indicating poten-tial for high-throughput applications. Experiments included measuring fluorescence changes upon UV irradiation, showcasing the capability to accurately quantify nucleic acids amidst interfering backgrounds, including tests in complex sample matrices like blood serum, proving the method’s practical applicability in real-world scenarios. This research advances the field of nucleic acid detection by introducing a novel photoswitch-ing technique, offering a powerful tool for clinical diagnostics and gene therapy.”
Comment 3: Minor issue about the section numbering. After 3.2 or 3.3, there should be 3.2.1 or 3.3.1. Now, they are all 3.1.x. Please revise.
Response: Thank you for your detailed review. We have corrected the numbering of the sections now.
Comment 4: Please check the figure captions. If a figure is from literature, please add references.
Response: We have thoroughly reviewed the figure captions. For figures sourced from the literature, we have added the appropriate references and have secured the necessary permissions for their use.

Reviewer 2 Report
Comments and Suggestions for Authors
Reviewer’s Report
Manuscript ID: molecules-3010110
Title: Recent Advances in Photoswitchable Fluorescent and Colorimetric Probes
Journal: Molecules
The aim of the manuscript "Recent Advances in Photoswitchable Fluorescent and Colorimetric Probes" is to comprehensively review and analyze recent developments in the field of photoswitchable fluorescent and colorimetric probes. The manuscript aims to provide valuable insights into the design, synthesis, and application of these probes by systematically presenting various types of molecular photoswitches and their photophysical properties, photoisomerization mechanisms, and fundamental design principles. Additionally, it analyzes the structural and electronic changes undergone by these probes upon light exposure, enabling them to switch between "on" and "off" states at specific wavelengths. Furthermore, it elaborates on the applications of photoswitchable probes in molecular detection, biological imaging, material science, and information storage, with a focus on their sensitivity and selectivity in detecting targeted analytes such as cations, anions, small molecules, and biomacromolecules. Ultimately, the manuscript aims to offer perspectives on the current state and future development of photoswitchable probes, guiding researchers in the design and application of new, efficient fluorescent and colorimetric probes, while serving as a clear introduction for researchers in the field.
This is a rather comprehensive review-type manuscript that gathers all relevant recent information regarding the use of photoswitchable fluorescent and colorimetric probes. I highly recommend that the Editorial office consider this manuscript for publication but after minor revision.
Reviewer’s suggestions
While the abstract mentions various applications of photoswitchable probes, such as molecular detection and biological imaging, it does not clearly define the scope of the review. A clearer outline of the topics covered in the manuscript should be provided.
Figures 2. b, c, and d are not visible well. Please, adjust the size of the figure. Check other figures as well. Maybe authors should consider decreasing the number of figures, just a suggestion.
Lines 111-114. This sentence has to be supported by relevant references
Lines 140-144. This sentence has to be supported by relevant references
Lines 174-177. This sentence has to be supported by relevant references
Lines 205-208. These sentences have to be supported by relevant references
Lines 241-243. This sentence has to be supported by relevant references
Lines 255-259. These sentences have to be supported by relevant references
Lines 292-293. This sentence has to be supported by relevant references, it is [83] and it should be Li et al.
Line 313. It should be Keyvan Rad et al.
Line 316 It should be Zhang et al.
Line 330 It should be Li et al.
Line 373 It should be Wang et al.
Line 389. It should be Ducrot et al.
Line 394. It should be Takaku et al.
Line 428. Something is wrong here, there are two Al signs.
Line 432. It should be Li et al.
Line 454. It should be Raman et al.
Line 454. It should be Cai et al.
I will no longer suggest correction regarding the citation proposition, the author should know that, if needed, they should cite in the body text of the manuscript the surname of the first author followed by et al. Please check this throughout the entire manuscript.
Lines 703 and 743. Just use the abbreviation WHO, in line 683 you have already defined WHO.
Author Response
Reviewers’ comments
Reviewer 2
The aim of the manuscript "Recent Advances in Photoswitchable Fluorescent and Colorimetric Probes" is to comprehensively review and analyze recent developments in the field of photoswitchable fluorescent and colorimetric probes. The manuscript aims to provide valuable insights into the design, synthesis, and application of these probes by systematically presenting various types of molecular photoswitches and their photophysical properties, photoisomerization mechanisms, and fundamental design principles. Additionally, it analyzes the structural and electronic changes undergone by these probes upon light exposure, enabling them to switch between "on" and "off" states at specific wavelengths. Furthermore, it elaborates on the applications of photoswitchable probes in molecular detection, biological imaging, material science, and information storage, with a focus on their sensitivity and selectivity in detecting targeted analytes such as cations, anions, small molecules, and biomacromolecules. Ultimately, the manuscript aims to offer perspectives on the current state and future development of photoswitchable probes, guiding researchers in the design and application of new, efficient fluorescent and colorimetric probes, while serving as a clear introduction for researchers in the field.
This is a rather comprehensive review-type manuscript that gathers all relevant recent information regarding the use of photoswitchable fluorescent and colorimetric probes. I highly recommend that the Editorial office consider this manuscript for publication but after minor revision.
Comment 1: While the abstract mentions various applications of photoswitchable probes, such as molecular detection and biological imaging, it does not clearly define the scope of the review. A clearer outline of the topics covered in the manuscript should be provided.
Response: Thank you for your valuable feedback and the recognition of our work. This review paper primarily focuses on photoswitchable fluorescent and colorimetric probes. These probes are capable of selectively responding to specific targets, where the detection and/or signal output involves a photoswitching process. Following your suggestion, we have supplemented the abstract to better define the scope of the review.
“This review systematically presents photoswitchable fluorescent and colorimetric probes built on various molecular photoswitches, primarily focusing on the types involving photoswitching in their detection and/or signal response processes.”
Comment 2: Figures 2. b, c, and d are not visible well. Please, adjust the size of the figure. Check other figures as well. Maybe authors should consider decreasing the number of figures, just a suggestion.
Response: Thank you for your comments. We have carefully checked the clarity of each figure and ensured that all images are clear. Additionally, we have re-uploaded high-resolution versions of all figures in the submission system. Considering the numerous examples cited in the latter part of the paper, we chose not to reduce the number of figures to maintain the readability of the article. Thank you for your valuable suggestion.
Comment 3: Lines 111-114. This sentence has to be supported by relevant references
Lines 140-144. This sentence has to be supported by relevant references
Lines 174-177. This sentence has to be supported by relevant references
Lines 205-208. These sentences have to be supported by relevant references
Lines 241-243. This sentence has to be supported by relevant references
Lines 255-259. These sentences have to be supported by relevant references
Response: Thank you for your professional advice. We have added the appropriate references at the specified lines to support the statements made.
Comment 4: Lines 292-293. This sentence has to be supported by relevant references, it is [83] and it should be Li et al.
Line 313. It should be Keyvan Rad et al.
Line 316 It should be Zhang et al.
Line 330 It should be Li et al.
Line 373 It should be Wang et al.
Line 389. It should be Ducrot et al.
Line 394. It should be Takaku et al.
Line 428. Something is wrong here, there are two Al signs.
Line 432. It should be Li et al.
Line 454. It should be Raman et al.
Line 454. It should be Cai et al.
I will no longer suggest correction regarding the citation proposition, the author should know that, if needed, they should cite in the body text of the manuscript the surname of the first author followed by et al. Please check this throughout the entire manuscript.
Response: Thank you for your comments. We have added the reference [83] ([89] in the revised manuscript due to the insertion of new references) and removed the duplicated aluminum symbol. Previously, we cited papers by the corresponding author’s surname throughout the text; following your suggestion, we have now revised these to reflect the first author’s surname.
Comment 5: Lines 703 and 743. Just use the abbreviation WHO, in line 683 you have already defined WHO.
Response: After initially defining WHO, we have used the abbreviation "WHO" directly in subsequent mentions throughout the manuscript.

Reviewer 3 Report
Comments and Suggestions for Authors
In this paper, the authors clearly and concisely described the current research progress of Photoswitchable Fluorescent and Colorimetric Probes. The structure of this paper is simple and clear: the authors firstly talked about the commonly used chemical structures of photoswitchable probes, then the authors described the applications areas of these probes of sensing cations, anions, small molecules, and biomacromolecules. lastly, the authors conclude the main topic and give their perspectives of this research field.
I recommend the publication of this review article without any further revision.
Author Response
Reviewers’ comments
Reviewer 3
In this paper, the authors clearly and concisely described the current research progress of Photoswitchable Fluorescent and Colorimetric Probes. The structure of this paper is simple and clear: the authors firstly talked about the commonly used chemical structures of photoswitchable probes, then the authors described the applications areas of these probes of sensing cations, anions, small molecules, and biomacromolecules. lastly, the authors conclude the main topic and give their perspectives of this research field.
Response: Thank you very much for your review and for the positive feedback on our manuscript. We have further revised and improved the document in the revised submission. We hope to have enhanced its quality. Thank you for your comments.